# Topographic and Biomechanical Changes after Application of Corneal Cross-Linking in Recurrent Keratoconus

**DOI:** 10.3390/ijerph16203872

**Published:** 2019-10-12

**Authors:** Emilio Pedrotti, Grazia Caldarella, Adriano Fasolo, Erika Bonacci, Nicola Gennaro, Alessandra De Gregorio, Giorgio Marchini

**Affiliations:** 1Department of Eye Clinic, Department of Neurosciences, Biomedicine and Movement Sciences, University of Verona, 37100 Verona, Italy; graziacaldarella@gmail.com (G.C.); adriano.fasolo@yahoo.it (A.F.); bonaccierika89@gmail.com (E.B.); giorgio.marchini@univr.it (G.M.); 2Epidemiology Unit, Veneto Region, 35100 Padua, Italy; nicola.gennaro@azero.veneto.it; 3Ophthalmic Unit, San Bassiano Hospital, 36061 Bassano del Grappa, Italy; adegre3@gmail.com

**Keywords:** recurrent keratoconus, corneal cross-linking, corneal ectasia, corvis

## Abstract

*Background*: Recurrent keratoconus (RKC) develops as a progressive thinning of the peripheral and the inferior cornea after keratoplasty, in both graft and host, causing secondary astigmatism, refractive instability, and reduced visual acuity. We evaluated the effectiveness of corneal cross-linking (CXL) in patients diagnosed with RKC. *Methods:* Accelerated-CXL via the epi-off technique was performed in15 patients (18 eyes) diagnosed with RKC. Topographic and biomechanical changes were assessed at 12 months. *Results*: Differences in maximum keratometry, thinnest corneal thickness, and biomechanical parameters (deformation amplituderatio, inverse concave radius, applanation 1 velocity, and applanation 2 velocity, stiffness A1) versus baseline were statistically significant (*p* < 0.05).Best corrected visual acuity was improved in 13 eyes and unchanged in 4;manifest refractive spherical equivalent was reduced in 13 eyes, increased in 3,and unchanged in 1 eye; topographic astigmatism was reduced in 9 eyes, remained stable in 1 eye, and increased in 7 eyes. *Conclusions*: Improved topographic and biomechanic indexes at 1 year after CXL suggest it‘s potential as first-line therapy for RKC, as it is for KC.

## 1. Introduction

Recurrent keratoconus (RKC) develops as a progressive thinning of the peripheral and the inferior cornea after keratoplasty. Corneal thinning occurs in both the graft and the host cornea, resulting in high secondary astigmatism, refractive instability, and reduced visual acuity [1].It is one of the main complications of keratoplasty, with a mean occurrence at17.9 years (range 7–40) and an estimated prevalence of 7%at 20 years and 12% at 25 years [2,3].

Currently, there is no consensus on the topographic criteria for RKC: An increase in maximum keratometry (Kmax) combined with a reduction in best-corrected visual acuity (BCVA) [4], and an increase in both steepest keratometry and cylinder [5] have been proposed as diagnostic criteria.

The first step in the diagnosis of RKC is topography to detect the increased curvature and secondary astigmatism, and then tomography to evaluate thinning of the ectatic cornea. But these processes cannot valuate corneal stiffness or mechanical stability. For these reasons, and for gaining a better understanding of the development of corneal ectasia, instruments that measure corneal biomechanics in vivo have attracted increasing interest.

Non-surgical management of RKC relies on the use of personalized contact lenses [6], but tolerance to wearing them may be lost as the disease progresses [7]. Surgical treatment includes laser in situ keratomileusis, astigmatic keratotomy (AK), and wedge resections [8,9]. Repeating penetrating keratoplasty (PK) at an increased diameter could also be an option, although approaching the limbus increases the risk for iatrogenic limbal stem cell damage and graft rejection [10,11]. These approaches yield unsatisfactory results, however, because they act on the graft cornea, for refractive purposes, and not on the host cornea, which is the site of disease.

Riboflavin/ultraviolet A corneal cross-linking (CXL) of corneal collagen has proven effective in halting progression of ectasias and deterioration of visual acuity in patients with progressive KC and post-refractive corneal ectasia [12,13], but little is known about the effectiveness of CXL on RKC. Richoz et al. reported that corneal CXL in three patients (three eyes) with RKC halted the progression of ectasias up to one year and suggested it as an additional treatment option for such patients [4]. The aim of the present study was to evaluate the feasibility and effectiveness of accelerated CXL in patients with RKC and transparent graft. 

## 2. Materials and Methods 

We enrolled consecutive patients with a history of keratoplasty performed for keratoconus and diagnosed with RKC. The study was conducted from September 2017 to February 2019at the Eye Clinic of the University of Verona in accordance with the tenets of the Declaration of Helsinki and the principles of good clinical practice. No approval of Ethics Committee was required for retrospective studies with anonymized data. Written, informed consent was obtained from all patients. Data collection and management were carried out in compliance with Italian data privacy law (Legislative decree 196 of 30 August 2003).

### 2.1. Ophthalmic Examination

For the purposes of this study, all patients were followed up for at least 10 years and evaluated with topography every 6 months. Diagnosis of RCK by topography (Eye Top, Costruzione, Strumenti Oftalmici, Scandici/Florence, IT) showed an increase in Kmax of at least 1 diopter (D) in the last 12 months, and changes in visual acuity.

Follow-up visits were scheduled at 4 and 8 days and at 1, 2, 3, 6, and 12 months after the procedure. Examination included slit lamp biomicroscopy, corneal pachymetry (Galilei G4, Ziemer Ophtalmic Systems, Port, CH), topography (Eye Top), biomechanical assessment (non-contact tonometer Corvis® ST, Oculus Optikgeräte GmbH, Wetzlar, GE) and BCVA (CSO Vision Chart, CSO) under controlled lighting conditions at 4 meters using a decimal scale (Snellen chart).

### 2.2. CXL Procedure and Care

CXL epi-off technique was performed after topical anesthesia in sterile conditions. The corneal epithelium was removed over the keratoplasty flap (approximately 1mm) in the host cornea using a blunt spatula, and then a solution containing riboflavin (0.1% riboflavin) was instilled for 15 min. A total of 9.0 mm of cornea was exposed to UVA light (power 9 mW/cm^2^) for 10 min (Q. Light Stativ, Q. Products, CH) [14]. A hyposmotic 0.5% riboflavin solution was imbibed in eyes with a corneal thickness < 440 μm. At the end of the procedure, a therapeutic contact lens was applied to the cornea. Post-operative therapy comprised: Netilmicin 3 mg/mL + dexamethasone 1 mg/mL eye drop four times a day for 10 days (tapered by one drop per day every 15 days thereafter), ophthalmic solution based on sodium, hyaluronate, amino acids, and vitamin B2 six times per-day, and oral supplementation of amino acids for 3 months. The therapeutic contact lens was removed when re-epithelialization was complete.

### 2.3. Biomechanics

Corneal assessment with the Corvis®-ST allows non-contact imaging acquired via a high-speed camera of dynamic corneal response (DCR) to an air pulse, and acquisition of data on corneal stiffness and viscoelastic properties, standard tonometry, and central corneal thickness (CCT). In detail, during the deformation response, a precisely metered air pulse causes the cornea to move inward or to flatten (first applanation). The cornea continues to move inward until it reaches a point of highest concavity. Because of the cornea’s viscoelastic nature, it rebounds from this concavity to another point of applanation (second applanation), and then back to its normal convex curvature [15].

The DCR parameters evaluated in this study were: Speed of corneal apex at first and second applanation (Appl1 and Appl2 velocity), distance between the two bending peaks created in the cornea at the maximum concavity state (highest concavity peak distance, HCPD), inverse concave radius (1/R), that represents the radius of curvature at highest concavity, deformation amplitude ratio (DA Ratio) at 1 or 2 millimeters from the center of cornea, stiffness parameter A1 (SP-A1), and biomechanical intraocular pressure (bIOP) compensating for CCT, age, and biomechanics.

### 2.4. Statistical Analyses

Results of descriptive analyses are expressed as means ±standard deviation (SD), median and range for quantitative variables, and as a count and percentage for categorical variables. All parameters were evaluated as changes within individual eyes. The main outcome measures to assess the effects of CXL were topographic Kmax, pachymetric thinnest corneal thickness (TCT), and DCR. Statistical differences in values of Kmax and TCT were evaluated by Steel Multiple Comparison Wilcoxon test (R statistical package, version 3.5.1). Bonferroni correction was applied for multiple comparisons. A paired-sample Wilcoxon test was performed to compare baseline values of DCR to those ofa 1-year follow up (STATA 13.0 statistical package (Stata Corp. LP, College Station, TX, USA).

A *p*-value < 0.05 was considered statistically significant. The distribution of the outcome parameters are presented as box-whiskers plots in the style of Tukey. 

## 3. Results

The study sample was 15 patients (18 eyes, four women and 11men, mean age 48.5 ± 9.2 years, range 34–61).Penetrating keratoplasty (PK) had been performed in 16 eyes and deep anterior lamellar keratoplasty (DALK) in two at 16.2 (5.5; 12–29) and 8.5 (12.1; 7–10)years before the diagnosis of RKC, respectively. A bilateral graft was present in three patients with previous PK.

The diameter of the donor graft was between 8.00 and 8.50 mm, and the diameter of host cornea between 7.75 and 8.25 mm. The residual stromal beds of the two eyes that had undergone DALK were approximately 100µm.All grafts were clear, and none of the patients reported ocular surface inflammation, corneal neovascularization or rejection episodes during the follow-up period after corneal transplantation. Five patients (six eyes) wore rigid gas permeable (RGP) contact lenses before enrollment in the study.

Complete closure of the corneal epithelium occurred within eight days after CXL. Corneal haze was noted up to the end of the first month, and complete recovery of corneal transparency by the end of the second month, after which combination corticosteroid and antibiotic eye drop therapy was suspended. Graft diastasis over 360° developed in one eye two days after the operation. The graft was sutured and the patient was withdrawn the study.

### 3.1. Topography and Pachymetry Results

Figure 1A shows the results of CXL on maximum keratometry (Kmax). Kmax was 64.49 D (8.67; 55.14–90.56) before surgery, 66.23 D (11.00; 54.37–90.07) at 1 month, 63.54 D (9.62; 51.04–87.93) at three months, 62.90 D (9.71; 51.18–87.39) at six months, and 61.63 D (9.13; 51.81–86.94) at one year after surgery. The differences in Kmax before and after CXL were statistically not significant using the Steel Multiple Comparison Wilcoxon test: *p* = 0.1305. After applying the Bonferroni correction differences in Kmax before and after CXL were statistically not significant at one month (*p* = 1.00) and three months (*p* = 0.32), significant at six months (*p* = 0.04), and 12 months (*p* < 0.004).

Figure 2 shows CXL topography results for a representative case in which Kmax was unchanged, whereas a reduction was noted in simulated keratometry dioptric mean of the steepest meridian (K1), simulated keratometry dioptric mean of the flattest meridian perpendicular to K1 (K2), topographic cylinder (Cyl), apical gradient curvature (AGC), average keratometry (AVG), and symmetry index (SI) at 1 year.

TCT was observed in the host cornea in proximity to the graft-host junction in all eyes.Figure 1B shows the results of CXL on TCT.TCT before surgery was 534 μm (64.38; 382–600), 546 μm (64.15; 385–612) at one month, 529 μm (65.84; 376–590) at three months, 530.57 μm (64.04; 367–587) at 6months, and 518.20 μm (66.54; 393–590) at one year. The difference in TCT before and after CXL was statistically not significant using the Steel Multiple Comparison Wilcoxon test: *p* =0.3475. After applying the Bonferroni correction difference in TCT before and after CXL was statistically not significant at one month (*p* = 0.16) and three months (*p* = 0.08), statistically significant six months (*p* = 0.04), and one year (*p* = 0.04).

### 3.2. Biomechanical Results

Deformation amplitude (HCDA) was reduced from 1.16 (0.10; 1.01–1.36) mm before surgery to 1.12 (1.10; 0.98–1.32) mm at 12 months; Appl1 velocity was reduced from 0.14 (0.03; 0.1–0.24) m/s before surgery to 0.13 (0.03; 0.1–0.22) m/s at 12 months; Appl2 velocity was increased from −0.28 (0.05; −0.21–(−0.44)) m/s before surgery to −0.26 (0.05; −0.19–(−0.42)) at 12 months; highest concavity radius (HCR) was increased from 6.10 (0.87; 7.79–4.48) mm to 6.16 (0.87; 7.83–4.52) mm−1.

Stiffness A1 (SP-A1) was increased from 51.28 (7.54; 34.8–63.3) mmHg/mm to 52.04 (7.65; 35.4–63.6) mmHg/mm.

Statistically significant differences were noted for the distribution of DA Ratio, 1/R, SP-A1, Appl1 velocity, and Appl2 velocity before and after CXL (*p* < 0.001) (Figure 3), whereas no statistically significant differences were noted in Appl 1 length, Appl 2 length, and HCPD at one year.

bIOP and CCT, as measured with the Corvis® ST tonometer, remained unchanged from baseline to 12-month evaluation: 13.27 mm Hg (2.0; 10.5–16.8) and 13.12 mm Hg (1.73; 10.3–16.2), 529.76 microns (34.39; 408–650) and 527.88 microns (63.20; 398–597), respectively.

### 3.3. Refraction Results

A reduction in visual acuity due to de-epithelialization and haze during the first week after CXL was recorded for all patients. A progressive increase in BCVA was noted at 12months in 13 eyes, with an improvement of three Snellen lines in three eyes, two Snellen lines in four, and one Snellen line in six. BCVA was unchanged in four.

Table 1 presents the refraction data. At one year, topographic astigmatism was reduced in 9eyes, increased in seven, and remained unchanged in one eye; the manifest refraction spherical equivalent (MRSE) was reduced in 13 eyes, increased in three, and remained unchanged in one eye.

The topographic cylinder value before CXL was 7.47 D (4.67; 2.5–20.94), 7.66 D (4.60; 3.32–21.89) at 1 month, 7.42 D (4.68; 2.91–21.75) at 3 months, 7.38 D (5.00; 3.10–23.56) at 6 months, and 7.02 D (4.76; 2.80–21.97) at one year. Optical correction with glasses were prescribed in 11 eyes, while the use of RGP contact lenses was continued in six.

### 3.4. Safety Evaluation

No infectious events were recorded during the follow-up period. Diastasis of the donor button over 360° developed in one patienttwo days after the procedure; he subsequently underwent graft suture and was withdrawn the study.

## 4. Discussion

Patients with previous keratoplasty (KP) for keratoconus (KC) and later diagnosed with recurrent KC (RKC) are treated by re-transplantation or other surgical options performed on the corneal graft for refractive purposes. The effectiveness of CXL in KC is amply documented, but the literature is scant on CXL for RKC.

Rabinowitz suggested that RKC results from the persistence of the peripheral corneal ring after KP, or from the incomplete removal of peripheral ectasia during surgery or due to new onset from the diseased peripheral stroma. Based on this assumption, RKC can be considered a pancorneal disease [16]. The presence of corneal ectasia in both the central and the peripheral host cornea demonstrates that KC is a pancorneal disease and that KP, even at large diameter, may not be enough to completely remove the disease from the host eye.

In our study, preoperative topography showed that ectasia recurred in the peripheral host cornea and then involved the graft cornea, demonstrating that RKC has its onset from the residual stromal ring. Although CXL performed over an 8-mm zone can effectively stabilize the peripheral cornea in KC [17], the ectasia onset in RKC starts from the peripheral cornea. Richoz performed epi-off CXL in patients with RKC, applying irradiation at 1 mm within the limbus. In this study, CXL over a 9-mm zone, including both the graft and the host cornea strengthened and stabilized the whole cornea, as confirmed by the topographic and biomechanical findings. The reduction in Kmax indicated not only a halting but also a regression of the ectatic process, while the changes in DCR parameters were correlated with stiffening of the corneal structure, with a significant decrease in DA Ratio, 1/R and Appl1 velocity at one year after CXL, and a significant increase in Appl2 velocity and SP-A1 at one year after CXL.

The reduction in DA Ratio, in 1/R and the increase in SP-A1 we noted at 12 months demonstrate an improvement in corneal stress resistance and stiffness after CXL. Furthermore 1/R and DA Ratio are useful parameters of DCR by their demonstrated independence on intraocular pression [18].

Values of bIOP and CCT measured by Corvis resulted unchanged from baseline to 12 months. It should be noted that the bIOP equation was not developed for patients with KC, and especially for those who had undergone keratoplasty. Moreover the equation assumes that the cornea’s age and patient’s age coincide, however, this is not factual for the eyes included in this study.

We found that the biomechanical behavior (both at diagnosis and after CXL) of RKC was similar to KC [19]. However, the lack of studies on RKC, none of which to our knowledge have assessed biomechanical behavior, precludes comparison with previous studies. 

We hypothesize that the reduction in TCT observed at one year stems from keratocyte apoptosis and compression and contraction of collagen fibers, as expected after CXL for KC. The literature reports no statistically significant reduction in TCT after accelerated CXL; however, these studies were conducted on KC and not on RKC [20,21].

## 5. Conclusions

An improvement in topographic and biomechanic indexes at one year after accelerated CXL for RKC was observed. These observations suggest CXL as first-line therapy for RKC, as it is for KC. Further studies including an appropriate control group and longer follow-up are desirable to determine whether CXL for RKC is effective to delay or obviate the need for KP.

## Figures and Tables

**Figure 1 ijerph-16-03872-f001:**
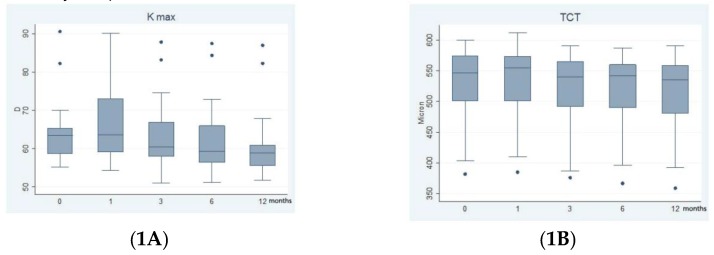
Effect of corneal cross-linking (CXL) on maximum keratometry (Kmax) and thinnest corneal thickness (TCT); (**1A**): Changes in Kmax were statistically significant at 6 months (*p* = 0.04) and 12 months (*p* = 0.004) versus baseline. (**1B**): Changes in TCT were statistically significant at 6 month (*p* = 0.04) and 12 months (*p* = 0.04) versus baseline.

**Figure 2 ijerph-16-03872-f002:**
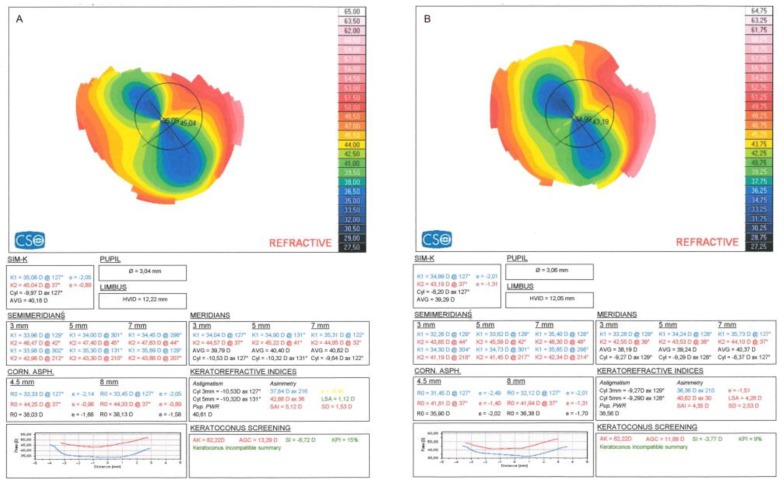
Corneal topography results before surgery (**A**) and at 12 months (**B**).

**Figure 3 ijerph-16-03872-f003:**
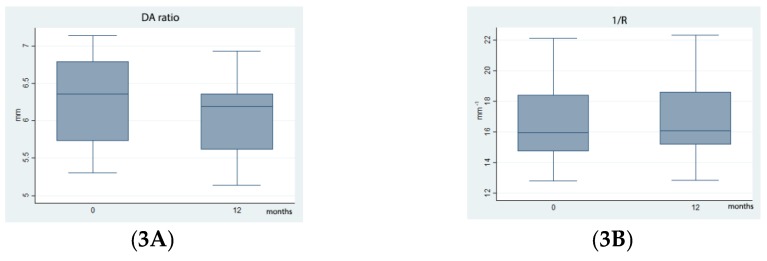
Biomechanical changes after CXL.

**Table 1 ijerph-16-03872-t001:** Topographic astigmatism and manifest refraction spherical equivalent (MRSE).

Eye	Topographic Astigmatism before CXL	Topographic Astigmatism at 1 year	MRSE before CXL	MRSE at 1 year
1	5.46	5.70	−1.25	−0.75
2	5.93	6.56	−5.25	−3.00
3	4.66	5.07	−4.75	−3.25
4	8.21	10.64	−4.50	−3.50
5	9.97	8.20	+6.50	+8
6	2.5	2.17	−4.25	−2.25
7	9.43	9.75	−10.50	−11.25
9	16.81	15.29	−6.00	−4.25
10	20.94	21.97	−4.00	−3.25
11	5.96	5.58	−2.25	−1.50
12	3.2	2.80	+0.50	+1.75
13	3.26	3.37	−3.50	−1.50
14	8.15	8.15	−5.00	−4.25
15	3.48	3.25	−1.25	−1.00
16	5.46	4.89	−5.25	−5.00
17	4.88	4.16	+3.25	+2.75
18	10.26	9.57	−1.75	−1.75

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
