# Peer review of "Topographic and Biomechanical Changes after Application of Corneal Cross-Linking in Recurrent Keratoconus"

_ijerph, 2019, doi:10.3390/ijerph16203872_

Round 1

Reviewer 1 Report

This manuscript have provided changes after application of corneal cross-linking in recurrent keratoconus. The comment are listed below : 1. Manuscript, especially introduction, results and discussions, should be edited. 2. As a minimum, the project identification code, date of approval and name of the ethics committee or institutional review board should be cited in the Methods section.

Author Response

Thank you for your comments.

All subjects gave their informed consent for inclusion before they participated in the study. The study was conducted in accordance with the Declaration of Helsinki and in according to good clinical practice. Was not required any approval of Ethics Commitee because it wasn’t conducted a clinical trial (you will find our corrections in the text: p.2, l. 62-64). 

We also edited introduction, results and discussion as you can find highlighted  in yellow in the text.

Reviewer 2 Report

Authors performed accelerated CXL on 18 eyes of recurrent keratoconus and evaluated corneal properties by topography and Corvis. They emphasized the importance of irradiating UV on host corneal bed.

As there are few reports on CXL on recurrent keratoconus, this study will be of interests of corneal surgeons.

My major concern is that how much the effect of CXL on recipient cornea was. Also there were some writing errors.

Major points.

p.1, l.35-37,

Increase of Maximum Keratometry (Kmax) in corneal topography of more than 1 Diopter, combined with a decrease in best-corrected visual acuity (BCVA) of at least 1 Snellen line over 1 year, is accepted as the diagnoses of RKC[4].

                I do not agree that there are accepted RKC diagnosing criteria. See Niziol et al, Am J Ophthalmol(2013), Pramanik S, et al, Ophthalmology(2006), Patel et al, Br J Ophthalmol(2009), and Yoshida J et al, Graefes Arch Clin Exp Ophthalmol (2018). They use different criteria for RKC. Details were discussed in Yoshida J et al. Graefes Arch Clin Exp Ophthalmol (2018)

p.2, l.67-68,

Diagnosis of RCK was made by topography (Eye Top, CSO, IT), showing increase of Kmax of at least 1D in the last 12 months and visual acuity instability.

                The topographic values vary to some extent after PK with high astigmatism but how did you determine the Kmax increase? How many times the exam performed to judge the Kmax increase?  

p.4, l.138,

Which are the thinnest corneal points were seen, donor or recipient cornea? In RKC cases usually recipient cornea is thinner.

p.4, Fig 1

In this figure they statistically compared only the baseline and 12 months by paired t-test. As this figure contains the repeated exams over time, non-parametric multiple comparison should be done to compare the baseline to each observation.

p.4, l.150, What does the figure 2 represent? Is this just an example of a representative case?

p.4, l.153, Kmax reduced => it did not change as authors wrote in the manuscript.

p.6-7, l.217-219,

CXL performed in the 9 mm of central cornea affected both the graft and the host stroma resulted in strengthening and stabilizing the whole cornea, a 218 s confirmed by topographic and biomechanical changes.

               Authors aimed to strengthen host cornea by CXL. However, as this study does not have controls (CXL within graft), they cannot say the topographic and mechanical changes were due to the CXL on host stroma.

p.7, l.232-

                Authors may want to discuss Corvis results in terms of the difference/similarity with KC cases without PK

p.7, l.237-

                TCT values reduced after CXL. As RKC is progressive condition, TCT should have reduced before CXL as well. How TCT reducing rate changed?

p.7, l.244-

the accelerated-CXL performed on RKC represented a viable alternative to regraft or other surgical procedures,

                This conclusion cannot be made because this study did not compare the CXL to the regraft or other surgery.

Writing errors

p.1 l.14, than =>and?

p.1., l.30, p.2, l.67 RCK => RKC? Unify writing.

p.3, l.126,  with drawn => was withdrawn

p.3, l.128, unchanges -> unchanged.  Spell out the acronyms since those are the first-time appearance

p.4, l.141 The title is incomplete.

Author Response

Thank you for your comments! You will find our correction in the text as indicated in brackerts.

There isn’t an unambiguous definition for RKC. We have not specified the behavior of astigmatism: all patients included in this study had a preoperative increase of cylinder and steepest keratometric power, also with an increase of Kmax and a decrease in best-corrected visual acuity (BCVA) according with the existing literature (p.1, l. 35-37). 

All patients were followed for at least  10 years and evaluated through topography every 6 months. We enroled patients showing an increase in Kmax also with a decrease in BCVA in the last 3 years (p.2, l. 66).

We found the thinnest corneal points in the host cornea in proximity of junction between graft and donor (p. 4, l. 137).

Non parametric multiple comparison was done to compare the baseline to each observation (p.3, l. 106-107).

We are sorry for our mistake in caption of Figure 2: it corresponds to a representative case, in particular to the only eye which had stability of Kmax after 12 months from CXL (p.4, l.156-160).

We were wrong to write the captation of figure2: Kmax is stable as showed in the figure and as described previously in the text (p. 3-4 l. 132-136).

We are sorry that come out wrong. We didn’t say that the effects of treatment are due to the strenghthening of the host cornea only, but for the streighthening of the whole cornea on which CXL was performed. We decided to perform CXL in the central 9 mm of cornea (including both host and graft) because RKC develops from host cornea and then involves the graft one. For these reasons we didn’t create a control group (CXL within graft) (p. 7-8, l. 221-225).

The biomechanical behaviour  in RKC and in KC is similar. All eyes showed increase of Def Amp, a decrease of Radius in highest concavity, Also after CXL the biomechanical behaviour is similar. CXL- RKC showed reduction in Def Amp and an increase of radius in highest concavity. We can not give detailed information on any significant difference between primary desease and recurrence because in this study we enrolled just RKC and we don’t had a controll group (p.8, l. 239-240).

TCT progressively reduced in RKC in the last 3 years as well as Kmax increased. We didn’t conduct a statistical analysis because it wasn’t the aim of our study.

We don’t have enough data to say for certain that CXL in RKC is a viable alternative to regraft, but the improve of topographic and biomechanic indexes demonstrated after 1 year can suggest CXL as a first-line therapy for RKC as for KC. Futher studies including appropriate controll group and longer term follow-up are usefull to conclude if CXL in RKC is effective to retard or avoid KP (p.8, l. 251-255).

Reviewer 3 Report

Interesting paper with significant novelties. 

the casuistry is numerous, the patients have been well evaluated and the follow-up is adequate

Appropriate some improvement and clarification

In “materials and methods” the period in which the patients were evaluated should be indicated.

Did all patients have the same postoperative therapy? what therapy was prescribed?

In the discussion it must be examined whether the different thickness of the residual stromal beds can have a meaning in the possibility of relapse of the KC or in the efficacy of the CXL treatment

A patient had a post-operative diastasis: did the patient have any particular characteristics that could make this complication fear?

It is known that CXL over an 8-mm zone can stabilize the peripheral cornea (Correlation of central and peripheral keratometric parameters after corneal collagen cross-linking in keratoconus patients Int Ophthalmol. 2018 Nov 12): do you think this data could be important in the proposed treatment? this should be discussed

Author Response

Thank you for you comments!

All patients were followed for al least 10 years and evaluated through topography every 6 months from keratoplasty. The study was conducted from September 2017 to February 2019 (p. 2, l. 61 and p.2, l. 66).

All patients recived the same post operative therapy as you can read in the text (p. 2, l. 81-86).

The topic put foward  on the role of the residual stroma is very interesting,  unfortunately the cohort of our patients includes only 2 eyes undergone to DALK so our experience is very limited and was not possible to assess the effects of presence of the stromal residue. The most important data that we obtained is the difference in time of development of RKC between eyes undergone to DALK and eyes undergone to PKP: 8.5 years mean and 16.2 mean respectively .

The patient who had a post-operative diastasis didn't have any particulary characteristic that could have predicted this complications.

It is known that CXL performed over an 8-mm zone can stabilize the peripheral cornea in keratoconus, but in literature there is very few knowledge about RKC. We decided to perform CXL in the central 9 mm of cornea (including both host and graft) because RKC develops from host cornea and then involves the graft one (p. 7-8, l. 221-225).

Round 2

Reviewer 1 Report

This manuscript have provided changes after application of corneal cross-linking in recurrent keratoconus.

Paragraphs should consist of more sentences( in the text: p.1. 62-64; p7. 234-235). Moreover, English language and style are minor spell check required (in the text: p.6. 62-64; RKC is considered a pancorneal disease. [16].). It should be improved.

Author Response

Thank you for your comments!

The study was conducted in accordance with the Declaration of Helsinki and in according to good clinical practice. No approval of Ethics Committee was required because it wasn’t conducted a clinical trial. All patients gave their informed written consent. Data collected were managed according to Italian legislative decree n° 196 of 30 August 2003 (p.2, l. 61-66).

Although other Authors had detected correlation between Def Amp, CCT and IOP [19], we did not found correlation of these parameters in our study: values of IOP and CCT measured by Corvis resulted unchanged from baseline to 12-month evaluation suggesting that our biomechanical results are not influenced by these parameters ( p. 9, l. 23-242).

Reviewer 2 Report

The Wilcoxon test is not suitable for "multiple" comparison in such a case. Steel test should be performed to compare between baseline and each time point.

Author Response

Thank you for your comment!

Statistical differences in values of Kmax and TCT were evaluated by Steel Multiple Comparison Wilcoxon test as you reccomended. Bonferroni correction was applied for multiple comparisons.  A paired-sample Wilcoxon test was performed to compare baseline values of DCR to those of 1-year follow up (p.4, l. 106-109).